# A synthetic dataset primer for the biobehavioural sciences to promote reproducibility and hypothesis generation

Daniel S Quintana*

Norwegian Centre for Mental Disorders Research (NORMENT), Division of Mental Health and Addiction, University of Oslo, and Oslo University Hospital, Oslo, Norway

**Abstract** Open research data provide considerable scientific, societal, and economic benefits. However, disclosure risks can sometimes limit the sharing of open data, especially in datasets that include sensitive details or information from individuals with rare disorders. This article introduces the concept of synthetic datasets, which is an emerging method originally developed to permit the sharing of confidential census data. Synthetic datasets mimic real datasets by preserving their statistical properties and the relationships between variables. Importantly, this method also reduces disclosure risk to essentially nil as no record in the synthetic dataset represents a real individual. This practical guide with accompanying R script enables biobehavioural researchers to create synthetic datasets and assess their utility via the *synthpop* R package. By sharing synthetic datasets that mimic original datasets that could not otherwise be made open, researchers can ensure the reproducibility of their results and facilitate data exploration while maintaining participant privacy.

**\*For correspondence:**
daniel.quintana@medisin.uio.no

**Competing interests:** The author declares that no competing interests exist.

## Introduction

Openly accessible biomedical research data provide enormous utility for science, society, and the economy (*Arzberger et al., 2004*; *Munafò et al., 2017*; *Murdoch and Detsky, 2013*; *Piwowar et al., 2011*). With open data, scholars can verify results, generate new knowledge, form new hypotheses, and reduce the unnecessary duplication of data collection (*Asendorpf et al., 2013*; *Nosek et al., 2012*). However, the benefits of data sharing need to be considered in light of disclosure risk. Researchers who wish to share data while reducing the risk of disclosure have traditionally used data anonymization procedures, in which explicit identifiers such as names, addresses, and national identity numbers are removed (*Hrynaszkiewicz et al., 2010*). To add additional disclosure protection, particularly sensitive variables (e.g., age) are sometimes aggregated and random noise may be added to the dataset. Despite these anonymization efforts, specific individuals can still be identified in anonymized datasets with high accuracy (*Ohm, 2009*; *Rocher et al., 2019*). Data aggregation and random noise can also distort the relationships between variables in the dataset (*Purdam and Elliot, 2007*), which can interfere with reproducibility and exploratory data analysis.

The creation of synthetic datasets can substantially overcome replicability issues, as this method creates a new dataset that mimics an original dataset by preserving its statistical properties and relationships between variables (*Little, 1993*; *Reiter, 2005b*; *Reiter, 2005a*; *Reiter and Raghunathan, 2007*; *Rubin, 1993*). Synthetic datasets also reduce disclosure risk to essentially zero, as no complete casewise record in the new dataset represents a real individual (*Duncan and Elliot, 2011*). Synthetic datasets also allow researchers to fit exploratory models in the synthetic datasets, which the data custodians can verify in the original data. Finally, synthetic datasets enable readers and

**eLife digest** It is becoming increasingly common for scientists to share their data with other researchers. This makes it possible to independently verify reported results, which increases trust in research. Sometimes it is not possible to share certain datasets because they include sensitive information about individuals. In psychology and medicine, scientists have tried to remove identifying information from datasets before sharing them by, for example, adding minor artificial errors. But, even when researchers take these steps, it may still be possible to identify individuals, and the introduction of artificial errors can make it harder to verify the original results.

One potential alternative to sharing sensitive data is to create 'synthetic datasets'. Synthetic datasets mimic original datasets by maintaining the statistical properties of the data but without matching the original recorded values. Synthetic datasets are already being used, for example, to share confidential census data. However, this approach is rarely used in other areas of research. Now, Daniel S. Quintana demonstrates how synthetic datasets can be used in psychology and medicine.

Three different datasets were studied to ensure that synthetic datasets performed well regardless of the type or size of the data. Quintana evaluated freely available software that could generate synthetic versions of these different datasets, which essentially removed any identifying information. The results obtained by analysing the synthetic datasets closely mimicked the original results.

These tools could allow researchers to verify each other's results more easily without jeopardizing the privacy of participants. This could encourage more collaboration, stimulate ideas for future research, and increase data sharing between research groups.

reviewers to better understand the data, as they can recreate the reported analyses and explore data distributions, variance, outliers, and means.

Synthetic datasets were originally developed for sharing sensitive population-level data (for a summary, see *Bonnéry et al., 2019*). The use of synthetic data for sharing sensitive information is beginning to emerge in the biobehavioural sciences (e.g., *Arslan et al., 2018*; *Newbury et al., 2018*); however, this approach is not widely known in the field. Given the benefits of synthetic data, the purpose of this article is to introduce this concept using examples and an accompanying R script. The R script and datasets to reproduce the analyses described in this paper are available online at https://github.com/dsquintana/synthpop-primer (*Quintana, 2019*; copy archived at https://github.com/elifesciences-publications/synthpop-primer). This website also includes a link to a RStudio Server instance of the primary analysis and results, which recreates the complete computational environment used for this manuscript (i.e., the R version and R package versions used) and facilitates straightforward reproducibility of the analysis described in this article via any modern web browser.

## Methods, materials, and results

Three open datasets will be used to demonstrate how the generation of a synthetic dataset can produce a dataset that mimics the original, via the *synthpop* R package (version 1.5–1; *Nowok et al., 2016*) in the R statistical environment (version 3.6.0). Synthetic datasets are created by replacing some or all of the data by sampling from an appropriate probability distribution, in a similar fashion to multiple imputation (*Drechsler, 2011*; *Raghunathan et al., 2003*; *Reiter, 2005b*). This approach preserves the statistical properties and the relationships between variables in the original dataset while safeguarding anonymity as no individual in the new dataset represents a real individual. The default synthesis method in *synthpop* is the classification and regression tree (CART) procedure (*Reiter, 2005b*), which will be used in this primer. The CART procedure is more flexible and generally performs better than other synthesis methods, such as random forests and bagging (*Drechsler and Reiter, 2011*). If data synthesis happens to recreate a replicate of a real individual by chance, these replicates can be easily identified and removed from the dataset to reduce the risk of de-identification.

There are two broad approaches for assessing the utility of a synthesized dataset: general utility and specific utility (*Snoke et al., 2018*). General utility reflects the overall similarities in the statistical properties and multivariate relationships between the synthetic and original datasets. The first step

in assessing general utility is data visualisation (*Nowok et al., 2016*; *Raab et al., 2017*). The compare() function in *synthpop* can be used to construct side-by-side univariate distributions of variables in the synthetic and observed datasets. Visualizing bivariate comparisons between specific variables of interest is also recommended, as two datasets might have similar statistical properties despite different distributions (e.g., Anscombe's quartet; *Anscombe, 1973*). Confirming general utility is a necessary step for making inferences from the synthetic dataset and is especially important for data exploration in the synthetic dataset (*Snoke et al., 2018*).

Specific utility for fitted synthetic models can be assessed by calculating the lack-of-fit against the same model in the original data (*Nowok et al., 2016*). Specific coefficients in synthetic models can also be compared against coefficients in the original models by assessing the differences in standardized coefficients and confidence interval (CI) overlap (*Karr et al., 2006*), which is the ratio of the overlap of the intervals to an average of their lengths. Higher overlap is indicative of greater specific utility, which suggests that inference from fitted synthetic models is valid.

Finally, there is a risk that individuals coming across the synthetic data without context (i.e., a direct link to a data file) may believe the data are real. Thus, to remove any possible confusion, synthetic datasets prepared for export will have a single string 'FAKE_DATA' variable added to the front of the dataset using the sdc() function in *synthpop*, as recommended by *Nowok et al. (2017)*.

## Example 1: Oxytocin and self-reported spirituality

*Van Cappellen et al. (2016)* investigated the impact of oxytocin administration on self-reported spirituality and deposited the raw study data online (https://osf.io/rk2x7/). In a between-participants design, volunteers were randomly assigned to self-administer an intranasal oxytocin (N = 41) or intranasal placebo spray (N = 42). Approximately forty minutes after receiving the nasal spray, participants were asked "Right now, would you say that spirituality is important in your life?". The reported outcome from an ANCOVA suggested that when accounting for religious affiliation, participants who self-administered the oxytocin nasal spray reported that spirituality was more important in their lives compared to those who self-administered the placebo spray. A synthetic version of the original dataset was created using the syn() function from the *synthpop* package. A comparison of the four main variables of interest revealed similar distributions between the synthetic and the original datasets and no individual extreme values (*Figure 1A*). There were also no replicated unique sets of values. A bivariate comparison of self-reported spirituality (*Figure 1—figure supplement 1*) and religious affiliation (*Figure 1—figure supplement 2*) between the nasal spray groups suggested that the counts between the synthesized and original datasets were similar. The relationship between age and self-reported spirituality was also similar between datasets (*Figure 1—figure supplement 3*). Altogether, these visualisations were indicative that the synthetic dataset has good general utility.

Nasal spray group differences in self-reported spirituality will be examined using both an independent samples Welch's *t*-test and a linear regression model equivalent, with the former required to assess specific utility. This analysis in the original dataset suggested no significant difference in spirituality ratings between the nasal spray groups [*t* = 1.14, 95% CI (−0.45, 1.63), p=0.26]. An equivalent linear regression model yielded the same outcome, as expected. Estimating this linear model in the synthesized dataset revealed the same p-value outcome (*t* = −1.12, p=0.26).

The lack-of-fit test comparing the models generated in the original and synthetic datasets was not statistically significant, $X^2$ (2)=0.01, p=0.995. This suggests that the method used for synthesis retained all the relationships between variables that influenced the model fitting parameters. The standardized difference of the nasal spray condition coefficient between the synthesized and the observed data for this *t*-test was 0.003, which was not statistically significant (p=0.998). A comparison of confidence intervals revealed 99.94% CI overlap between the synthetic and original datasets (*Figure 1B*). Overall, these results indicate that the model from the synthesized data demonstrates high specific utility for this *t*-test. This particular outcome was not reported in the original article; however, this provides a demonstration of how synthetic data can be used for data exploration. The analysis script underlying exploratory analysis can be shared with the owners of the original dataset, for the easy verification of the analysis. In this case, applying this model to the synthetic data produces almost precisely the same results as the original dataset.

Next, let's explore the correlation between age and self-reported spirituality. A Pearson correlation test revealed no statistically significant correlation between age and self-reported spirituality

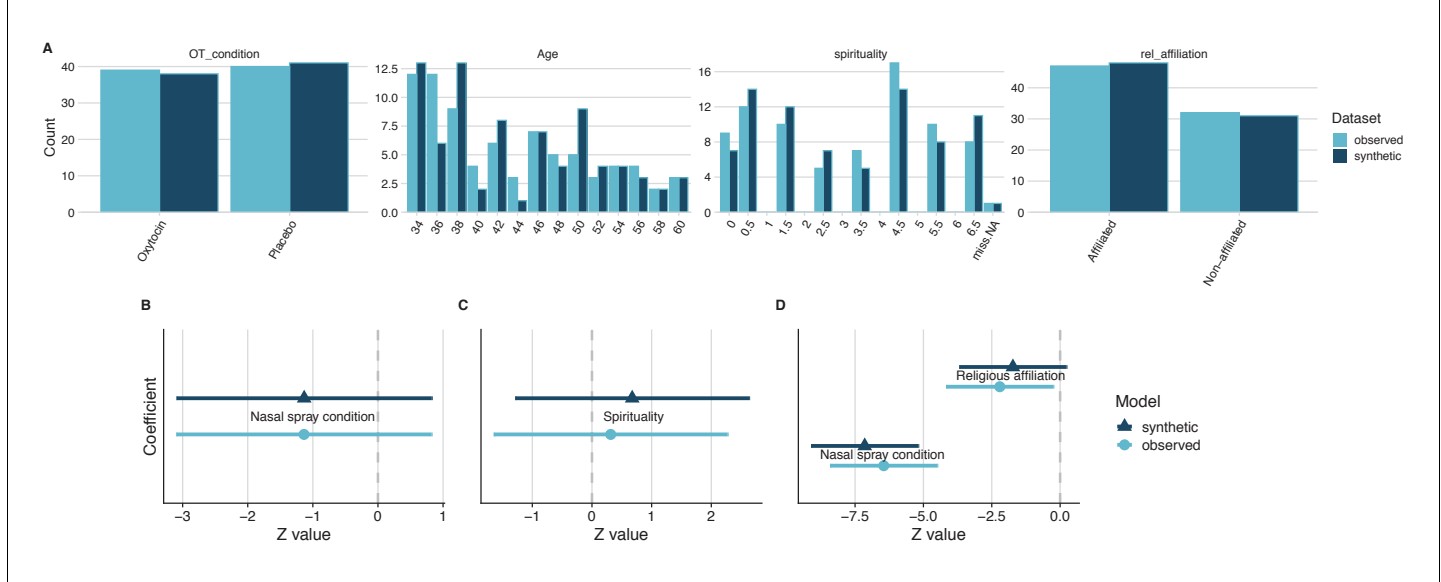

**Figure 1.** General and specific utility of synthetic data from a study on the impact of intranasal oxytocin on self-reported spirituality. A comparison of the four variables of interest revealed similar distributions in both the observed and the synthetic datasets, which is indicative of good general utility (**A**). Direct comparisons of coefficient estimates and 95% confidence intervals from linear models calculated from synthetic and observed datasets revealed no significant differences and high confidence interval overlap (**B–D**), which is indicative of good specific utility.

The online version of this article includes the following figure supplement(s) for figure 1:

**Figure supplement 1.** Differences in self-reported spirituality, stratified by nasal spray condition and dataset.
**Figure supplement 2.** Differences in religious affiliation, stratified by nasal spray condition and dataset.
**Figure supplement 3.** The relationship between age and self-reported spirituality in the observed and synthetic datasets.

[$r$ = 0.04, 95% CI (−0.19, 0.26), p=0.75], which is the same result as the linear model equivalent. Estimating this model using the synthetic dataset revealed a similar Pearson's $r$ value and a non-significant result ($r$ = 0.08, p=0.5). The lack-of-fit test comparing the models generated in the original and synthetic datasets was not statistically significant, $X^2$ (2)=0.13, p=0.94. Moreover, the test of the standardized differences between the synthetic and original data for the relationship coefficient was not statistically significant (p=0.72) and there was 90.8% CI overlap between the synthetic and observed data (**Figure 1C**), suggesting high specific utility.

Finally, let's explore the main reported outcomes from this study, that oxytocin increased self-reported spirituality controlling for religious affiliation. First, the original outcome was verified [F (1,75)=4.87, p=0.03] and then the analysis was structured as a linear regression model. This analysis yielded the same p-value of .03, which was associated with a t-statistic of −2.2. Estimating this model using the synthetic dataset revealed similar t-statistic and a p-value that was on the border of statistical significance (t = −1.8, p=0.07).

The lack-of-fit test for the full model was not statistically significant, $X^2$ (3)=0.91, p=0.82. The test of the standardized differences between the synthetic and original data for the nasal spray coefficient was not statistically significant (p=0.63) and there was 87.8% overlap between the synthetic and observed data (**Figure 1D**). Although the synthetic model nasal spray coefficient was not statistically significant, like the original model, this matters little given the considerable overlap between the confidence intervals of the synthetic and original models. It is worth a reminder at this point that just because one coefficient is significant and the other is not, this does not necessarily mean that these coefficients are significantly different from each other (**Gelman and Stern, 2006**). The primary interest in the comparison of synthetic and original models is effect size estimation and confidence interval overlap. The test of the standardized differences between the synthetic and original data for the religious affiliation coefficient was not statistically significant (p=0.49) and there was 82.3% overlap between the synthetic and observed data (**Figure 1D**). Overall, these results are indicative of high specific utility for the overall synthesized model and its coefficients.

## Example 2: Sociosexual orientation and self-rated attractiveness

Sociosexual orientation is described as an individual's propensity to participate in uncommitted sexual relationships. Given the personal nature of sociosexual orientation, this a good example of the type of sensitive data that individuals may be hesitant to share in some cases, thus demonstrating the benefit of releasing of synthetic data. *Jones and DeBruine, 2019* collected sociosexual orientation data from 9627 individuals using a revised version of the sociosexual orientation inventory (*Penke and Asendorpf, 2008*) both in the laboratory and online, along with data on self-rated attractiveness and basic demographic information. Fourteen variables were synthesized from the original dataset, which has been archived online (https://osf.io/6bk3w/). The synthetic data demonstrated good general utility, as the distributions of variables were comparable between the original and synthetic datasets (*Figure 2A*). A model was fitted to examine if self-rated attractiveness, data collection location (laboratory or online), and age predicted the number of times someone has had sexual intercourse on only a single occasion with another individual. Both self-rated attractiveness ($t = 13.64$, $p<0.001$) and age ($t = 27.69$, $p<0.001$) were statistically significant predictors, whereas location was not a significant predictor ($t = 0.05$, $p=0.96$). Estimating this linear model in the synthesized dataset revealed relatively similar $t$-statistic outcomes (self-rated attractiveness: $t = 14$, $p>0.001$; age: $t = 27.51$, $p<0.001$; location: $t = 1.12$, $p=0.26$).

The lack-of-fit test was not statistically significant, [$X^2 (4)=1.5$, $p=0.83$], suggesting that the method used for synthesis retained all the relationships between variables that influenced the parameters of the fit. The standardized difference of the self-rated attractiveness coefficient between the synthesized and the observed data was 0.32, which was not statistically significant ($p=0.75$). A comparison of confidence intervals revealed 91.8% CI overlap between the synthetic and original datasets (*Figure 2B*). The standardized difference of the age coefficient between the synthesized and the observed data for this $t$-test was $-0.28$, which was not statistically significant ($p=0.78$). A comparison of confidence intervals revealed 92.9% CI overlap between the synthetic and original datasets (*Figure 2B*). The standardized difference of the location coefficient between the synthesized and the observed data for this $t$-test was 1.08, which was not statistically significant ($p=0.28$). A

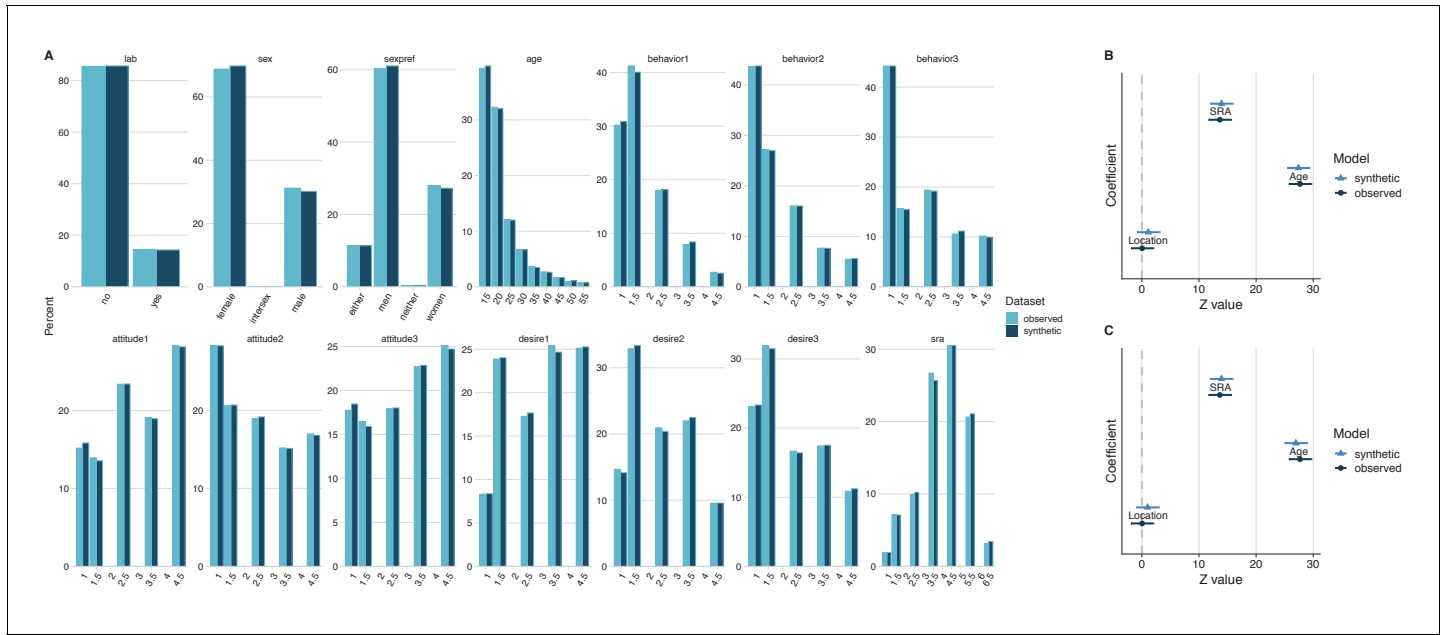

**Figure 2.** General and specific utility of synthetic data from an investigation on sociosexual orientation. A comparison of the fourteen variables of interest revealed similar distributions in both the observed and the synthetic datasets, which is indicative of good general utility (**A**). Direct comparisons of coefficient estimates and 95% confidence intervals from a linear model calculated from synthetic and observed datasets revealed no significant differences and high confidence interval overlap (**B**), which is indicative of good specific utility. The coefficient estimates and 95% confidence intervals of the same model derived from the synthetic dataset with 213 replicated individuals removed also demonstrated high confidence interval overlap (**C**). This demonstrates that reducing disclosure risk has little effect on specific utility.

comparison of confidence intervals revealed 72.4% CI overlap between the synthetic and original datasets (*Figure 2B*). Overall, these results indicate that this model from the synthesized data demonstrated high specific utility.

A comparison of the original and synthetic datasets revealed 213 replicated individuals (2.3% of the total sample). To reduce disclosure risk, these replicated values were removed and models were refitted to examine the effects of removal on outcomes. The lack-of-fit test between the original model and synthetic model with replicated individuals was not statistically significant, $X^2$ (4)=2.8, p=0.6. The refitted model coefficients yielded similar results to the model derived from the original data (self-rated attractiveness: standardized difference = 0.3, p=0.77, CI overlap = 92.4%; age: standardized difference = −0.74, p=0.46, CI overlap = 81.2%; location: standardized difference = 0.95, p=0.34, CI overlap = 75.8%; *Figure 2C*). Therefore, reducing the risk of disclosure by removing replicated individuals in the synthetic dataset maintains specific utility, in this case.

## Example 3: Heart rate variability and fitness levels in a series of simulated datasets

Heart Rate Variability (HRV) is a non-invasive measure of autonomic cardiac control (*Akselrod et al., 1981*) and thought to be positively correlated with fitness level (*Dixon et al., 1992*). The Root Mean Square of Successive Differences (RMSSD) is a commonly used HRV measure, however, its distribution tends to be positively skewed (e.g., *Kobayashi et al., 2012*). Moreover, missing data are common in HRV investigations due to equipment malfunction. To demonstrate the effects of the distribution pattern of HRV (normal, low skew, high skew), and missing data (none, 5%, 20%) for a range of sample sizes (40, 100, 10000), twenty-seven data sets with four variables (heart rate, weight, fitness level, and HRV) were simulated for the creation of synthetic datasets, which included outliers (*Supplementary file 1*). For all datasets, HRV and fitness level were modelled to have a relationship that is typically associated with a medium effect size (r = 0.3).

The specific utility of synthetic datasets was examined by comparing the relationship between HRV and fitness in each synthetic dataset to its respective original dataset. None of the lack-of-fit tests were statistically significant (*Supplementary file 1*), suggesting that the method used for synthesis retained all the relationships between variables that influenced the parameters of the fit. A comparison of confidence intervals revealed strong overlap between the synthetic and original datasets for most (but not all) of the 27 analyses and none of the standardized coefficient differences between the synthesized and the observed datasets were statistically significant (all p's > 0.05; *Figure 3*, *Figure 3—figure supplements 1–2*). However, the overlap between the synthetic and original models were on the border of statistical significance when synthesizing data with a low skew in the simulated samples with 10,000 cases (*Supplementary file 1*; *Figure 3—figure supplement 2*). All 27 synthetic datasets also demonstrated good general utility, regardless of the parameters (*Figure 3—figure supplements 3–5*). Thus, synthetic dataset generation in *synthpop* seems to be relatively robust against differences in sample size, missingness, and skew in these simulated samples, however, there were indications of poorer performance in some of larger datasets with 10,000 (*Supplementary file 1*). Altogether, it is crucial that general and specific utility is assessed for each synthesised dataset, as it is difficult to predict *before* synthesis how well the procedure will perform.

## Discussion

Researchers need to consider the trade-off between the risk of identification and the utility of open datasets when deciding whether to make their research data openly available. Open data can provide substantial utility, but this may expose research participants to the risk of identification. Conversely, closed datasets decrease the risk of disclosure to essentially zero, but have almost no public utility. The generation of synthetic datasets provides an appealing compromise, as synthetic data can offer comparable levels of utility as the original datasets while substantially reducing disclosure risk. The adoption of synthetic datasets in the biobehavioural sciences will improve reproducibility and secondary data exploration, as it will facilitate the sharing of data that would otherwise not be made available.

Study participants are generally in favour of researchers sharing their deidentified data (*Ludman et al., 2010*; *Mello et al., 2018*). Thus, when planning future studies researchers should include data sharing provisions when receiving participant consent (*Taichman et al., 2016*).

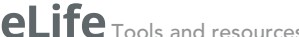

**Figure 3.** Specific utility of synthetic data from a range of simulated datasets with 100 cases that model the relationship between Heart Rate Variability (HRV) and fitness. Nine datasets with 100 cases were simulated, which varied on skewness for the HRV variable (none, low, high) and missingness for all variables (0%, 5%, 20%). The x-axes values represent Z-values for the HRV coefficient. The dark-blue triangles and confidence intervals represent the HRV estimates for the synthetic data and the light-blue circles and confidence intervals represent the HRV estimates for the observed data. In general, there was a high overlap between the synthetic and original estimates (*Supplementary file 1*). The confidence interval range overlap between the synthetic and observed estimates from the dataset with normally distributed HRV and 5% missing data were 60.5%. While the standardized difference was not statistically significant (p=0.12), caution would be warranted in terms of specific utility in this case, given the relatively low confidence interval range overlap.

The online version of this article includes the following figure supplement(s) for figure 3:

**Figure supplement 1.** Specific utility of synthetic data from a range of simulated datasets with 40 cases that model the relationship between Heart Rate Variability (HRV) and fitness.

**Figure supplement 2.** Specific utility of synthetic data from a range of simulated datasets with 10,000 cases that model the relationship between Heart Rate Variability (HRV) and fitness.

**Figure supplement 3.** General utility of nine simulated datasets with 40 cases.

**Figure supplement 4.** General utility of nine simulated datasets with 100 cases.

**Figure supplement 5.** General utility of nine simulated datasets with 10,000 cases.

Obtaining updated consent to share data from participants who have not provided this when originally participating in a study can be resource intensive. Some have suggested that sharing deidentified datafiles should not require new re-consent from participants (*Taichman et al., 2017*), but as mentioned above, many commonly used data deidentification approaches may not sufficiently reduce disclosure risk (*Ohm, 2009*; *Rocher et al., 2019*). In some circumstances, datasets may include extremely sensitive information that is difficult to anonymise. For instance, *Arslan et al. (2018)* collected highly sensitive data examining the role of ovulatory changes on sexual desire and behavior in women but did not request consent from participants to share data considering valid privacy concerns. Instead, a synthetic version of the dataset was created using *synthpop* and made

available on a publicly accessible repository. Releasing this synthetic dataset provides considerable utility, as other researchers can verify the analysis and fit novel models using this dataset, which can be confirmed by the data custodians with the original data. An additional step of removing individuals that have been fully replicated in the synthesized data set can further reduce disclosure risk, without necessarily reducing general or specific utility. Therefore, synthetic data can offer a valuable solution for sharing data collected under conditions where participants did not specifically provide consent for sharing data (and where re-consent is impractical), as well as for situations in which a dataset contains especially sensitive information. In addition to the verification of results and hypothesis generation, synthetic datasets can also benefit the training of machine learning algorithms in research areas with a dearth of data, such as rare condition research (*Ekbatani et al., 2017*; *Sabay et al., 2018*), via the creation of additional synthetic datasets that closely match real datasets.

One criticism of sharing raw data is that research groups would not have the first opportunity to analyse the data and report outcomes to their satisfaction (*Lo, 2015*; *Ross et al., 2012*). It has been recommended that secondary data analysts should seek collaborations with teams that collected the original data in recognition of their investment in collecting the data (*Taichman et al., 2016*), but this is difficult to enforce in practice. To make meaningful inferences with synthetic data, secondary data analysts need to verify their synthetic models against models from the original data, which is in the possession of the original authors who can verify these inferences (*Reiter et al., 2009*). This would increase the likelihood of co-authored collaborations, at least compared to the status-quo in which secondary analysts could publish their results without necessarily collaborating with the original authors. Thus, open synthetic data provide an opportunity for secondary analysists scholars to fit models that the original authors may not have considered, while also encouraging them to collaborate with the original authors to verify their models in the real dataset. Of course, secondary analysts could still report results from synthetic datasets without verification from the primary authors, but it would need to be made explicit that analyses were conducted on a synthetic dataset, and generated models may not necessarily mirror the models generated from the original dataset.

Journals have adopted a spectrum of public data archiving (PDA) policies, ranging from the policies that data should be made "available upon request" all the way to mandated data deposition in peer-reviewed journals dedicated to open data (*Sholler et al., 2019*). While an "available upon request" PDA policy is better than no policy at all (*Stodden et al., 2018*), such datasets are often difficult to retrieve in practice as corresponding authors can become unreachable or original datasets are lost (*Couture et al., 2018*; *Stodden et al., 2018*; *Wicherts et al., 2006*). Sharing data with published papers would remove these impediments for accessing data, with synthetic data offering a solution for when it is not possible to share the original dataset due to disclosure concerns.

Despite the benefits of synthetic datasets, this approach is not without limitations. First, it is possible for synthetic data to have poor general and specific utility, which would diminish the benefits of sharing in terms of reproducibility and secondary data exploration. While a synthetic dataset with poor utility would still provide a valuable guide for reproducing reported analyses, these are likely to provide substantially different estimates and exploratory analyses may not produce accurate models. Second, current synthetic data methods limit the types of statistical inference that can be performed on synthetic data to linear models. In practice, this means that only linear models can be for comparison in order to demonstrate specific utility. Of course, scholars are free to perform any type of analysis on the synthetic data, which should provide approximately the same outcome as the original data as long as the synthetic data offer good general utility. Third, as mentioned above, the risk of identity disclosure from synthetic datasets is negligible but this only holds under two conditions: that none of the complete synthetic data records match with the original data and that there are no extreme single individual values in the dataset that can be linked to an individual (*Drechsler, 2011*; *Duncan and Elliot, 2011*). Therefore, to reduce disclosure risk and the possibility that participants will recognise themselves in the dataset, complete matching records (i.e., when all variables in the original dataset for a case matches a case in the synthetic dataset) should be identified and removed. Additionally, in the case of categorical variables with only a few observations, scholars should consider collapsing these into another category (e.g., if there are only a few observations in an age band of 70–79 years old, this can be collapsed into the previous age band of 60–69 years old). If there are uncommon continuous values above or below a certain threshold, it may be prudent to collapse these into another category or creating a new category (e.g., top-coding a new '70+' age variable for any age equal to or above 70). While recoding may lead to synthetic datasets with

less utility, this approach might be required to reduce disclosure risk, something that data synthesizers will have to carefully consider in light of the sensitivity of the dataset along with national laws and guidelines.

When the creation of synthetic datasets for disclosure control was first proposed in the early 1990s, it was considered "rather radical" at the time (pg. 461; *Rubin, 1993*). Researchers have continued improving this method since these initial proposals (*Reiter, 2005b*; *Reiter, 2005a*; *Reiter and Raghunathan, 2007*), but only more recently has an easy-to-implement tool for creating synthetic data become available. The *synthpop* R package enables researchers to generate and share synthetic datasets that mimic original datasets with sensitive information. Importantly, the use synthetic data will improve the reproducibility of biobehavioral research and help generate novel hypotheses for future research (*Bonnéry et al., 2019*).

## Acknowledgements

This research was supported by an Excellence Grant from the Novo Nordisk Foundation to DSQ (NNF16OC0019856). The author would like to thank Steve Haroz, who provided helpful feedback on an earlier version of this manuscript.

## Additional information

### Funding

| Funder | Grant reference number | Author |
| --- | --- | --- |
| Novo Nordisk Foundation | Excellence grant NNF16OC0019856 | Daniel S Quintana |

The funders had no role in study design, data collection and interpretation, or the decision to submit the work for publication.

### Author contributions

Daniel S Quintana, Conceptualization, Data curation, Formal analysis, Funding acquisition, Investigation, Visualization, Methodology

### Author ORCIDs

Daniel S Quintana https://orcid.org/0000-0003-2876-0004

### Decision letter and Author response

Decision letter https://doi.org/10.7554/eLife.53275.sa1
Author response https://doi.org/10.7554/eLife.53275.sa2

## Additional files

### Supplementary files

- Supplementary file 1. Specific utility of simulated datasets.
- Transparent reporting form

### Data availability

Data and analysis scripts are available at the article's Open Science Framework webpage https://osf.io/z524n/.

The following previously published datasets were used:

| Author(s) | Year | Dataset title | Dataset URL | Database and Identifier |
| --- | --- | --- | --- | --- |
| Van Cappellen P, Way BM, Isgett SF, | 2016 | Effects of oxytocin administration on spirituality and emotional | https://osf.io/rk2x7/ | Open Science Framework, https:// |

| Fredrickson BL | | | responses to meditation | | osf.io/rk2x7/ |
|---|---|---|---|---|---|
| Jones BC, DeBruine L | | 2019 | Sociosexuality and self-rated attractiveness | https://osf.io/6bk3w/ | Open Science Framework, DOI: 10. 17605/OSF.IO/6BK3W |

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
