## [Decision Letter]

**Acceptance summary:**

We feel that the study is important as it touches on an emerging area of data sharing, provenance and reproducibility – a nascent movement towards achieving rigor and transparency that is being adopted by funding agencies, journal editors and industry alike. Applying the concept of synthetic datasets, in this case using SynthpopR, is seen as innovative and applicable to a wide range of investigations, importantly to those where full data and identity disclosure are not possible. The tool is likely to be utilized widely in translational and clinical research.

**Decision letter after peer review:**

Thank you for submitting your article "Synthetic datasets: A primer for the biobehavioural sciences to promote reproducibility and hypothesis-generation" for consideration by *eLife*. Your article has been reviewed by Christian Büchel as the Senior Editor, Mone Zaidi as the Reviewing Editor, and two reviewers. The following individual involved in review of your submission has agreed to reveal their identity: Dorothy VM Bishop (Reviewer #1).

The reviewers have discussed the reviews with one another and the Reviewing Editor has drafted this decision to help you prepare a revised submission.

Summary:

Creation of synthetic datasets has been adopted in some areas of social science where large survey datasets are used, and where full anonymisation is difficult or impossible. Here, the author demonstrates the usefulness of the synthpop R package for generating synthetic data in a different field, where datasets are typically smaller and simpler.

While the paper does not add new empirical observations or analysis approaches, it is, as its title indicates, a primer for others who might wish to try this approach. It highlights the benefits of synthetic datasets as compared to open research data. Importantly, the use of synthetic datasets shows reliable general utility in terms of data reproducibility of original findings. As long as the risk of identity disclosure is eliminated, synthetic datasets can and should be employed given the reproducibility of scientific data in biomedical sciences is a growing issue. The inclusion of oxytocin data re-analyzed in the synthetic datasets is a strength. The data are strong and well presented. The manuscript is clearly written for the most part and delivers a straightforward message that would be useful to a general readership. Nonetheless several areas need improvement and further analysis.

Essential revisions:

1) It would be preferable to have more diversity in the illustrative datasets – perhaps even demonstrating a case when synthpop does not perform as well. In particular, questions were raised regarding data with unusual distributions, evident outliers, or a hierarchical structure, for example, those with a code for family, and then family members.

2) There is mention in the Discussion section about cases where the synthetic dataset has poor utility. Perhaps, this cannot be predicted in advance, but it would be interesting to test this with datasets that may have the kinds of characteristics that are noted in point 1 (above). Indeed, this would all get rather meta, but one could simulate data that has specific features, such as striking outliers or non-normal distributions, to determine how far synthpop could retain its utility in such situations. Maybe this would be a better use of the second demonstration (see point 3). Would it be possible to take the original dataset and throw in some outliers or transform distributions to push synthpop to determine how well it stands up to non-normal data and the likes.

3) The three illustrative cases seem repetitive. Examples 1 and 3 show how the package worked with relatively large and small datasets, respectively, but example 2 seemed unnecessary. It is suggested that this part of an online appendix or is utilized it as noted in point 2 (above).

4) There is the issue about discarding data points that happen to replicate a real individual by chance. It is mentioned in the Introduction, again in Results section and in the Discussion section. A reviewer isn't sure about the logic of removing these. How would the individual – or anyone else – know this was their data? There is a problem with de-identification in a real dataset. For example, one knows that someone is a 55-year-old, left-handed woman, and there is only one of those in the sample. But if one knows that a dataset is synthesized, one could not be confident that the remainder of the data in that row came from this person.

5) It would be critical to model missing data. This might have an impact on fidelity of the synthetic data.

---

## [Author Response]

Essential revisions:1) It would be preferable to have more diversity in the illustrative datasets – perhaps even demonstrating a case when synthpop does not perform as well. In particular, questions were raised regarding data with unusual distributions, evident outliers, or a hierarchical structure, for example, those with a code for family, and then family members.

To increase diversity of the illustrative datasets, I have now removed one of the datasets (example 2) and included a series of simulated datasets instead (subsection “Example 2: Sociosexual orientation and self-rated attractiveness”). There are several approaches for how to simulate data to demonstrate synthpop’s robustness (or where it performs suboptimally). To best represent common characteristics of unusual datasets, I have simulated datasets that differ in sample size, skew, and the percentage of missing data. Outliers are also included among the simulated datasets.

These simulated datasets are modelled off common cardiovascular and demographic variables (heart rate variability, heart rate, fitness level, weight). In total, 3 groups (n = 40, n = 100, n = 10,000) of datasets have been simulated, and each group contains 9 datasets with the following characteristics:

Normal distribution and no missing data

Normal distribution and 5% missing data

Normal distribution and 20% missing data

Low skew and no missing data

Low skew and 5% missing data

Low skew and no missing data

High skew and no missing data

High skew and 5% missing data

High skew and no missing data

Thus, a total of 27 datasets are presented in the new illustrative example.

2) There is mention in the Discussion section about cases where the synthetic dataset has poor utility. Perhaps, this cannot be predicted in advance, but it would be interesting to test this with datasets that may have the kinds of characteristics that are noted in point 1 (above). Indeed, this would all get rather meta, but one could simulate data that has specific features, such as striking outliers or non-normal distributions, to determine how far synthpop could retain its utility in such situations. Maybe this would be a better use of the second demonstration (see point 3). Would it be possible to take the original dataset and throw in some outliers or transform distributions to push synthpop to determine how well it stands up to non-normal data and the likes.

As mentioned above I have created a new illustrative example with 27 simulated datasets. Instead of transforming the original dataset, I created a set of simulated datasets from scratch for more fine-grained control of the parameters. For each dataset, I present general and specific utility diagnostics. These simulations demonstrate that synthpop is remarkably robust in most cases to non-normal data and missing data, at least for these examples. However, the utility of simulated datasets is suboptimal for some of the samples. For example, the model generated for the synthetic version of the dataset with 100 cases, HRV data with a low skew, and 5% missing data overall only had a 60.5% confidence interval overlap with the model generated from the observed data. While the standardized difference between these two estimates was not significantly different, one should not place too much confidence in the synthetic model given the relatively modest confidence interval overlap, as this is indicative of weak specific utility. As suggested by the reviewer, it is difficult to predict which datasets will perform poorly a priori based on the dataset characteristics alone. Thus, it is crucial that each generated synthetic dataset is checked for general and specific utility.

3) The three illustrative cases seem repetitive. Examples 1 and 3 show how the package worked with relatively large and small datasets, respectively, but example 2 seemed unnecessary. It is suggested that this part of an online appendix or is utilized it as noted in point 2 (above).

As mentioned above, this example has been replaced by a set of simulated datasets.

4) There is the issue about discarding data points that happen to replicate a real individual by chance. It is mentioned in the Introduction, again in Results section and in the Discussion section. A reviewer isn't sure about the logic of removing these. How would the individual – or anyone else – know this was their data? There is a problem with de-identification in a real dataset. For example, one knows that someone is a 55-year-old, left-handed woman, and there is only one of those in the sample. But if one knows that a dataset is synthesized, one could not be confident that the remainder of the data in that row came from this person.

To clarify, my intention was to highlight the potential issue of the replication of cases in which all variables (rather than some variables) happen to replicate a real case. I have now made this clearer in the manuscript, as follows:

In the Introduction:

“Synthetic datasets also reduce disclosure risk to essentially zero, as no complete casewise record in the new dataset represents a real individual (Duncan et al., 2011).”

And in the Discussion section:

“Therefore, to reduce disclosure risk and the possibility that participants will recognise themselves in the dataset, complete matching records (i.e., when all variables in the original dataset for a case matches a case in the synthetic dataset) should be identified and removed”

5) It would be critical to model missing data. This might have an impact on fidelity of the synthetic data.

As mentioned above, I have now modelled the effects of missing data on the creation of synthetic datasets. The degree of missingness (none, 5%, or 20%) seems to make little systematic difference in terms of general or specific utility.